# Corneal Neurotization, Recent Progress, and Future Perspectives

**DOI:** 10.3390/biomedicines13040961

**Published:** 2025-04-14

**Authors:** Ovidiu Samoilă, Lăcrămioara Samoilă, Lorina Petrescu

**Affiliations:** 1Ophthalmology Department, University of Medicine and Pharmacy Iuliu Hatieganu, 400347 Cluj-Napoca, Romania; ciprian.samoila@umfcluj.ro; 2Physiology Department, University of Medicine and Pharmacy Iuliu Hatieganu, 400347 Cluj-Napoca, Romania; 3Vedis Ophthalmology Clinic, 400371 Cluj-Napoca, Romania; petresculorina@gmail.com

**Keywords:** corneal neurotization, neurotrophic keratopathy, corneal ulcer, reinnervation

## Abstract

Neurotrophic keratopathy (NK) is a rare degenerative disease caused by impairment of the trigeminal nerve, leading to corneal anesthesia, epithelial breakdown, and progressive vision loss. Conventional treatments primarily focus on symptom management and the prevention of complications, but they do not address the underlying nerve dysfunction. Corneal neurotization (NT) has emerged as a promising surgical intervention aimed at restoring corneal sensation and improving ocular surface homeostasis. This review evaluates the outcomes of corneal neurotization in patients with NK and compares the effectiveness of direct (DNT) and indirect (INT) techniques. Studies have reported significant improvements in corneal sensitivity, with success rates ranging from 60.7% to 100% (mean: 90%). Most patients experienced recovery of corneal sensation, as measured by the Cochet–Bonnet aesthesiometer, with no significant differences in outcomes between DNT and INT. Indirect neurotization using a sural nerve graft was the most commonly employed technique (63% of cases), while the use of acellular allografts demonstrated comparable efficacy and simplified the procedure. Postoperative corneal sensitivity increased significantly, from a preoperative average of 2.717 mm to 36.01 mm, with reinnervation typically occurring within 4–6 months and peaking at 12 months. In vivo confocal microscopy confirmed the presence of nerve regeneration. Neurotization was found to be safe, with minimal donor-site complications, which generally resolved within one year. Although the procedure improves corneal sensation and tear film stability, visual acuity outcomes remain variable due to pre-existing corneal damage. Early intervention is, therefore, recommended to prevent irreversible scarring. However, the number of patients undergoing the procedure remains limited, making it difficult to draw definitive conclusions. Most available studies consist of small case series. Further research with larger sample sizes is needed to refine surgical techniques and optimize patient selection, thereby improving outcomes in the management of NK.

## 1. Introduction

The cornea is the most sensitive organ in the human body, primarily due to the ophthalmic branch of the trigeminal nerve, which forms a dense network of nerve fibers across the corneal surface. Corneal innervation is essential for maintaining ocular surface homeostasis. It not only plays a critical role in the blinking and lacrimal reflexes but also contributes to the release of various trophic factors necessary for epithelial cell proliferation. Consequently, any alteration in corneal innervation leads to epithelial breakdown and the development of neurotrophic keratopathy (NK).

NK is considered a rare disorder, with an average prevalence of 5 per 10,000 individuals. It is caused by trigeminal nerve injury along any portion of its pathway—preganglionic, postganglionic, or at the ocular surface. The etiology of NK is diverse, including both ocular and systemic causes such as herpetic viral infections, malignancies, surgical trauma, and corneal injuries [1].

In current clinical practice, treatment for NK remains primarily conventional, aiming to prevent further corneal degradation without addressing the underlying nerve dysfunction. As a result, the disease is often only partially controlled. Recent studies have explored novel therapeutic options such as nerve growth factors. Cenegermin, a recombinant human nerve growth factor, has been approved for the treatment of NK. Although it has demonstrated efficacy in promoting epithelial healing, its extremely high cost limits widespread clinical use. Some studies suggest that it may also improve corneal sensitivity, but further interventional research is needed to validate these findings [2,3,4].

Corneal neurotization remains the only intervention capable of restoring corneal innervation and re-establishing ocular surface homeostasis. First described by Samii in 1972, it was initially deemed impractical due to its complexity and limited clinical benefit. However, recent years have seen the development of several modified techniques aimed at making the procedure more accessible in clinical practice. In 2009, Terzis introduced the concept of direct neurotization [5], and in 2014, Borchert’s group [6] implemented an indirect technique using a sural nerve graft coapted end-to-side to the contralateral supratrochlear nerve.

### Surgical Procedure

Corneal neurotization involves creating a connection between the denervated cornea and functional branches of the ipsilateral or contralateral trigeminal nerve. Several variations of the procedure currently exist, broadly categorized into direct neurotization (DNT) and indirect neurotization (INT). These range from more invasive open approaches to minimally invasive endoscopic techniques, all with the same goal: restoring corneal sensation.

The DNT technique involves transposing the ipsilateral or contralateral supratrochlear or supraorbital nerves directly to the anesthetic cornea. INT, by contrast, uses an interposed nerve graft—commonly the sural nerve—to bridge the gap between the donor nerve (typically the supraorbital or supratrochlear) and the denervated cornea. Other graft options, such as the greater auricular nerve or acellular nerve grafts, have also been reported [7,8,9,10]. For INT, a graft length of 10–15 cm is generally required [11].

Despite its complexity, corneal neurotization is a promising intervention that demands a multidisciplinary approach and involves a steep learning curve. Its growing success and continued refinement are driving increased interest in this technique.

The aim of this scoping review is to consolidate the scientific literature published in the past decade, systematize key concepts related to corneal neurotization, and compare surgical outcomes across different techniques. We conducted a systematic search for relevant articles, with inclusion limited only by the unavailability of full-text access. Compared to previous reviews—for example, Park et al. [12], who included 54 eyes—our review encompasses a larger patient population and proposes several directions for future research.

## 2. Materials and Methods

In this review, we examined recently published data on surgical techniques and clinical outcomes of corneal neurotization performed by surgeons with significant experience in the procedure. Therefore, only publications that included at least three patients were considered for inclusion.

The review was conducted following PRISMA guidelines. Studies were identified through searches on PubMed and Google Scholar using the keywords “corneal neurotization” and “corneal reinnervation” (last search date: January 2025; reviewers: S.O. and P.L.; both reviewers conducted independent research to ensure a low indexing bias; no formal tool was used for the assessment of study quality or other risk of bias). A total of 301 results were found, dating back to 1961. However, most of the relevant reports were published after 2017.

Inclusion criteria were as follows:Publication date between 2017 and 2024;Full-text availability;Articles written in English or French;Studies involving human subjects;A minimum of three patients per study.

Abstracts, conference communications, and other articles without full-text access were excluded (Figure 1). A total of 188 reports were assessed at the abstract level, and 170 were excluded for various reasons, including studies not limited to humans, data limited to experimental or preclinical findings, or misclassification by the database.

Eighteen studies were selected for full-text review. Of these, six included only one or two patients and were subsequently excluded from the final analysis.

Systematic errors in study selection were minimized by applying a limited set of exclusion criteria. Specifically, we focused solely on recent data and ensured that each included study involved an adequate number of human participants. 

We observed the demographic parameters—age, sex; surgical procedures—direct and indirect neurotization; anatomical outcomes—corneal healing, in vivo confocal microscopy (IVCM); and functional outcomes—central corneal sensibility (CCS), best corrected visual acuity (BCVA). Anatomical and functional outcomes were compared between studies, where available. Statistical tests for mean comparison were analyzed (with *p* value set at <0.05). Statistical analysis was not possible for the comparison of the studies, due to the lack of population homogeneity and the low number of patients. 

## 3. Results

Twelve studies met the inclusion criteria and were included in the review; they were published between 2018 and 2023 (Table 1). All patients were diagnosed with neurotrophic keratopathy (NK) and had reduced or abolished corneal sensation as measured by the Cochet–Bonnet aesthesiometer (CBA) before surgery, except for one study [13] in which corneal sensitivity was tested only qualitatively. The Cochet–Bonnet aesthesiometer determines central corneal sensation on a scale from 0 mm (no sensation) to 60 mm (full sensation).

The most reported causes of neurotrophic keratopathy were trigeminal lesions secondary to tumors, intracranial interventions, craniocerebral injuries, herpetic keratitis, and congenital causes, which were primarily described in children. Less frequently, neurotrophic keratopathy was secondary to repeated eye interventions or was idiopathic.

A total of 164 eyes underwent neurotization. Most cases (127 eyes) received indirect neurotization (INT). The sural nerve (or other grafts, such as acellular nerve allografts [9,10]) was attached to the supratrochlear or supraorbital nerve (and, less frequently, to the infraorbital nerve), using an end-to-end procedure (mostly) or end-to-side (35% of cases, as reported by Sweeney et al. [9]). A schematic of the intervention is shown in Figure 2. Thirty-seven cases received direct neurotization (DNT), where the contralateral intact supraorbital or supratrochlear nerve was directed to the damaged cornea (ipsilateral nerves may also be used if they are intact, as observed in the studies by Catapano et al. and Lin et al. [14,16]).

In both methods, the nerve graft was tunneled into the conjunctival space of the affected eye. The fascicles were separated and attached either perilimbal or into the peripheral corneal stroma via a corneoscleral tunnel incision. Multiple fascicles (usually 4 to 6) were implanted around the cornea in all quadrants. A temporal central tarsorrhaphy was sometimes necessary. In some cases, an amniotic membrane was applied either over the cornea or wrapping the anastomosis, based on the argument that it may assist with nerve regeneration due to the high levels of neurotrophic growth factors [20].

Neurotization is a complicated procedure. Elalfy et al. [19] estimated a surgical time of 5 h for the very first procedures performed by the team, which was gradually reduced to 3 h once two surgeons performed the sural nerve harvest and eye preparation procedures simultaneously. They also observed improvements in ocular surface and tear film stability (break-up time, Schirmer test, corneal staining). Only five patients were analyzed with in vivo confocal microscopy (IVCM), with nerve parameters improving at 3 months but showing no further improvement up to 12 months. Subjective corneal sensation appeared as early as 1.5 months and as late as 18 months in one patient.

Fogagnolo et al. [17] compared direct neurotization (DNT) (16 cases) to indirect neurotization (INT) (10 cases) in a multicenter prospective study. No differences in central corneal sensitivity (CCS) were observed between the groups at 12 months. A faster recovery time was seen in DNT, with CCS being statistically better at 3 and 6 months. All NKs healed after a mean period of 3.9 months (range, 2–6 months). However, three patients did not regain corneal sensitivity. Subbasal corneal plexus (SCP) was normal at 12 months on IVCM. While no major complications were observed, all patients who underwent DNT had mild facial edema, and all with INT had edema of the upper third of the face. Numbness at the harvesting site subsided after 12 months. Misperception of corneal sensitivity at the harvesting site was common in the first 3–6 months after surgery.

Catapano et al. [14] investigated mostly pediatric cases (mean age, 12.5 ± 8.3). Congenital corneal anesthesia was the primary cause of NK, and 26% of patients developed amblyopia. They also investigated the difference in corneal sensibility when nerve fascicles were inserted perilimbal versus through corneoscleral tunnels, observing a better, though not statistically significant, outcome in the latter group (2.5 ± 6.1 mm versus 19.4 ± 23.1 mm, *p* < 0.08). A total of 87% of patients achieved ≥40 mm central corneal sensitivity, and 64% achieved full sensitivity. Four patients developed persistent epithelial defects (PED), but only after penetrating keratoplasty (PK), all of whom had maximum CCS. PEDs were successfully treated in 4 weeks with lubricants and a bandage contact lens. Woo et al. [21] also included pediatric cases (17 patients out of 23). CCS improved to near normal in more than 60% of the eyes. The first sensation appeared in 3 months, and maximum sensation was achieved at an average of 11.1 months. Younger patients experienced more recurrent epithelial breakdown (mean age, 10.9), with the authors suggesting reduced awareness and an increased propensity for injury in children.

Weis et al. [13] claim to be the first study to present a series of adult patients with neurotrophic keratopathy (NT). All si patients regained corneal sensation within the first 6 months. All ulcers healed, and the epithelium remained stable.

Anatomical improvements were also assessed with IVCM, focusing on corneal nerve density. In Gianacarre et al. [15], the corneal sub-basal nerve plexus (SNP) was absent preoperatively, appeared at 3 months, and became comparable to a normal eye by 12 months. IVCM quantified several parameters, including corneal nerve branch density, corneal nerve fiber length, corneal nerve total branch density, corneal nerve fiber area, corneal nerve fiber width, and corneal nerve fractal dimension. All parameters improved, starting from the 3-month scan, and all but two parameters (corneal nerve fiber width and fractal dimension) showed progressive increases until the end of the 12-month follow-up. The study also reported significant improvements in tear film function following corneal neurotization, as evidenced by the Schirmer test and tear break-up time (TBUT). The mean Schirmer test score increased from 3.0 mm/5 min (±0.81) preoperatively to 7.66 mm/5 min (±1.24) postoperatively, while the TBUT improved from 2.6 s (±0.47) to 6.0 s (±0.81).

Leyngold et al. [10] observed peripheral improvement in corneal sensibility in all seven eyes, while central improvement was reported in five. The focus of this study, however, was the successful use of processed acellular nerve allograft (Avance Nerve Graft, a processed human nerve allograft intended for the surgical repair of peripheral nerve discontinuities). Acellular allograft seems to have outcomes similar to autograft and additionally simplifies the procedure [9,10]. However, more experimental studies with larger sample sizes are needed. Sweeney et al. [9] did not notice any differences in CCS between end-to-end coaptations (11 cases) versus end-to-side (six cases) coaptation techniques (52 mm versus 60 mm maximum CCS).

Wisely et al. [18] reported the first successful corneal neurotization performed simultaneously with PK. Corneal sensation appeared at 4 months (corneal irritation), and an objective improvement was noticed at 4.5 months.

All patients in Lin et al.’s [16] study had postherpetic NK. The procedure was well tolerated, with no intraoperative complications. The best-corrected visual acuity remained stable, dependent on corneal scarring typically seen in NK caused by ocular injuries, with further improvements expected after PK.

Su et al. [22] observed full epithelial recovery at 12 months and nerve regeneration at 6 months on average, with maximum regeneration occurring between 12 and 18 months. However, they observed a time difference between corneal nerve recovery and corneal sensation recovery. Corneal nerve fiber density recovered earlier (at 12 months) than corneal nerve branch density (at 18 months), with the latter being more strongly correlated (*p* < 0.05) with corneal sensation recovery. They evaluated and compared the recovery of central and peripheral sub-basal corneal nerve fiber density (CNFD) following NT surgery. While both regions experienced significant improvements over 24 months, no statistically significant differences were observed between the central and peripheral CNFD values (*p* > 0.05). Most, but not all, patients recovered corneal sensation, though the exact numbers were not specified.

## 4. Discussion

Almost all the patients included in the review had improvement in corneal sensation, as assessed by the Cochet–Bonnet aesthesiometer (Figure 3). The success rate for both DNT and INT can be estimated at about 90% (ranging between 60.7% and 100%). Studies describing both variants of neurotization have found no significant difference in the postoperative outcomes between the two groups (Fogagnolo et al. [17]).

Indirect neurotization with a sural nerve graft was the preferred procedure, applied to 63% of all eyes. Acellular allograft could replace the sural nerve and seem to have the same outcomes as autograft, while also simplifying the procedure (Sweeney et al., Leyngold et al., 24 eyes in total [9,10]). However, more experimental studies with larger sample sizes are needed.

The cause of NK did not seem to influence the outcome of the surgery in terms of corneal sensation and stability. Pediatric cases with congenital anesthesia of the cornea, as well as adult cases with causes ranging from intracranial surgeries to herpetic keratitis, all had similar outcomes [14]. Visual acuity did not always improve but usually remained at least stable, with no further deterioration. The variability in visual acuity was mainly correlated with corneal scarring in long-lasting NK, with improvements expected after PK. Therefore, corneal neurotization should be performed in the early stages of neurotrophic keratopathy before such changes occur. Amblyopia was another concern in young children. Catapano et al. [14] added other factors for the variability, including patient selection, surgical technique, disease duration, and the structural integrity of the cornea.

Preoperative CCS on CBA was 2.7 mm (averaged from the 10 articles reporting the parameter; range, 0–9.2 mm) and significantly improved after surgery to 36.01 mm (range, 21.1–49.7 mm). Corneal sensation appeared within 4.12 months on average (seven studies reported this parameter accurately; range, 3–6 months), and reached its maximum at 12 months (except for Sweeney et al., who reported maximum sensitivity at 6.6 months).

Catapano et al. [14] observed that cases with fascicles inserted into corneoscleral tunnels had better corneal sensation at early follow-up compared to those where fascicles were tunneled into the subconjunctival space. Woo et al. [21] suggested that chronic denervation and the use of a contralateral donor may cause delayed recovery after neurotization, but at the last follow-up, no significant differences were observed compared to the control group. They also found that the number of fascicles and insertions was negatively correlated with the results of reinnervation. The number of fascicles used in the studies included here ranged from 4 to 8.

Some studies used in vivo confocal microscopy (IVCM) to analyze corneal nerves. These studies showed significant improvement in nerve parameters after neurotization [10,16,17,18,19]. Su et al. [22] identified a temporal difference between the recovery of corneal nerve density and corneal sensation, highlighting the correlation between these factors. By analyzing patients with significant recovery at 24 months postoperatively, it was found that central and peripheral corneal nerve fiber density (CNFD) showed substantial recovery by 12 months, with continued improvement in peripheral nerves due to their thinner nature, as evidenced by IVCM images. Objective evidence of corneal sensation and reinnervation after neurotization could also be provided by magnetoencephalography (MEG) [14].

After corneal neurotization, there were cases where epithelial healing was present but corneal sensitivity had not yet returned. This was explained by the fact that nerve growth factors could provide protective trophic support to the cornea, independent of corneal sensation [17,19].

Giannaccare et al. [15] demonstrated increased tear film stability (as measured by the Schirmer test and TBUT), highlighting the overall enhancement in ocular surface health and tear film stability post-surgery. Lacrimal secretion was tested, and a significant increase in tear production was noted after corneal neurotization [20].

Neurotization was a safe procedure, with no intraoperative complications reported. Numbness at the donor site usually subsided within 1 year [10,18,19]. One study reported misperception of sensitivity at the harvested site for the first 3–6 months [19].

The pediatric population was included in the studies by Catapano et al. [14] and Woo et al. [21]. Younger patients experienced more recurrent epithelial breakdown, with the authors suggesting reduced awareness and an increased propensity for injury in children. However, nerve function was improved in proportions similar to those seen in adult studies, with over 60% of eyes reaching normal sensitivity. The maximum CCS was 49.7 mm and 44.1 mm, respectively (Figure 3).

### 4.1. Future Perspectives

Corneal neurotization appears to be a well-established procedure for the loss of corneal sensibility. With no serious complications and a high success rate, the procedure is safe and could become the first choice of treatment for NK. However, we still lack sufficient data, as relatively few patients have received treatment thus far. The procedure is relatively new, with a more detailed description emerging only in the last 15 years. Some refinements to the procedure already exist, including the possibility of replacing the sural nerve with an acellular nerve in INT or the increased use of amniotic membrane at the corneal site. Most of the studies examined here were case series (ranging from as few as 3 eyes to no more than 28 eyes), with larger cohorts currently unavailable. 

Further research should also focus on the differences between adult and pediatric populations. Pediatric patients may have better regenerative capacity and may show more rapid sensory recovery following corneal neurotization, as seen in Catapano et al. [14], where children showed substantial improvement in CCS. However, younger patients experienced more epithelial breakdown (propensity to injury or reduced awareness)**.** Adult patients, especially those with acquired conditions like trauma, may have slower or less complete recovery due to aging nerve tissues and reduced regenerative capacity. Future studies should stratify outcomes by age to better understand these differences. 

Outcome measurement is another challenge that surgeons face. IVCM is not easily accessible, and the CBA has some degree of subjectivity. Pain, ultimately, is a subjective experience, so improvements in sensitivity scoring may benefit only the scientific community and not the patient. Standardized outcome reporting across studies is crucial to ensure comparability and consistency in assessing the effectiveness of corneal neurotization. Additionally, long-term neurophysiological assessments could provide insights into the durability of nerve regeneration and functional recovery over time. We suggest that the presence of nerve filaments tunneled around the cornea could be observed using optical coherence tomography (OCT) (Figure 4). Refinements in anterior segment OCT could improve nerve fiber detection at the level of the corneal sub-basal plexus.

This review also highlights the need for more robust patient-reported outcomes (PROs), including quality of life and symptom relief, to better understand the patient-centered benefits of the procedure. Finally, randomized controlled trials (RCTs) with larger sample sizes and longer follow-up periods would provide higher-quality evidence regarding the efficacy of corneal neurotization for NK.

### 4.2. Limitations

As mentioned earlier, there are not enough patients who have received neurotization to date. Only 164 patients in total were included in this review. Some of the articles included lacked data regarding CCS or objective measurements, such as IVCM. It seems that only a few centers have experience with a large number of patients, with most of the reports found in online databases being case reports or small sample studies.

## 5. Conclusions

Corneal neurotization is an excellent option for addressing the refractory cases of neurotrophic keratopathy, gaining more and more ground in recent years due to stable results over time.

## Figures and Tables

**Figure 1 biomedicines-13-00961-f001:**
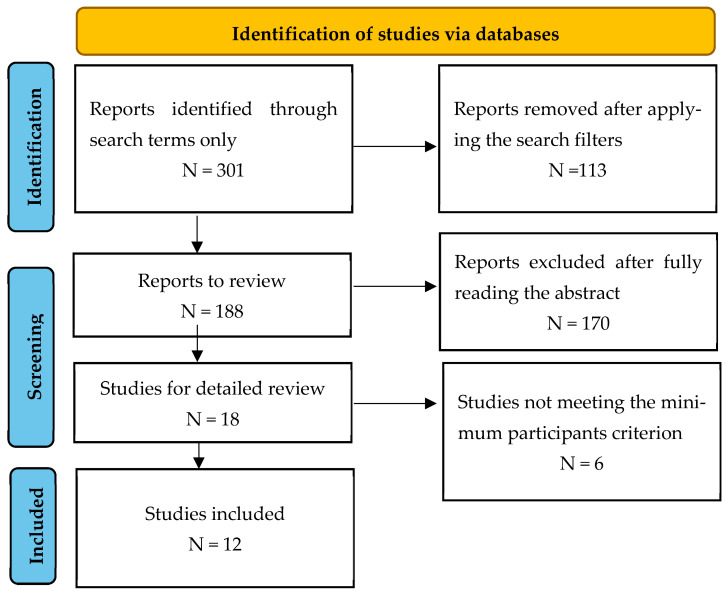
PRISMA flow diagram.

**Figure 2 biomedicines-13-00961-f002:**
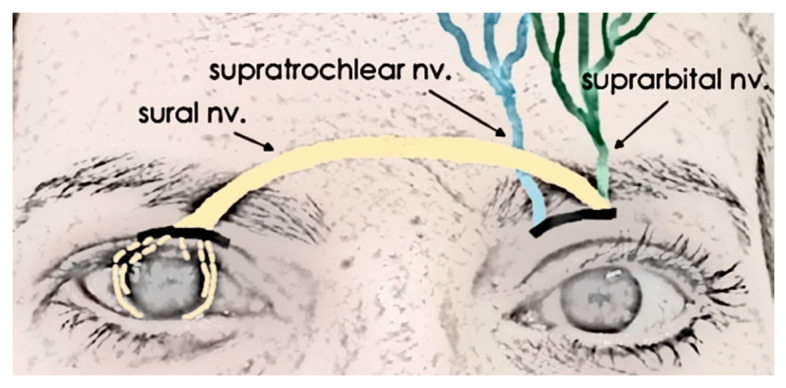
Indirect neurotization (INT), showing sural nerve grafting to the supraorbital nerve. The same graft could be applied to the supratrochlear nerve. Multiple fascicles are inserted at the 4 corneal quadrants.

**Figure 3 biomedicines-13-00961-f003:**
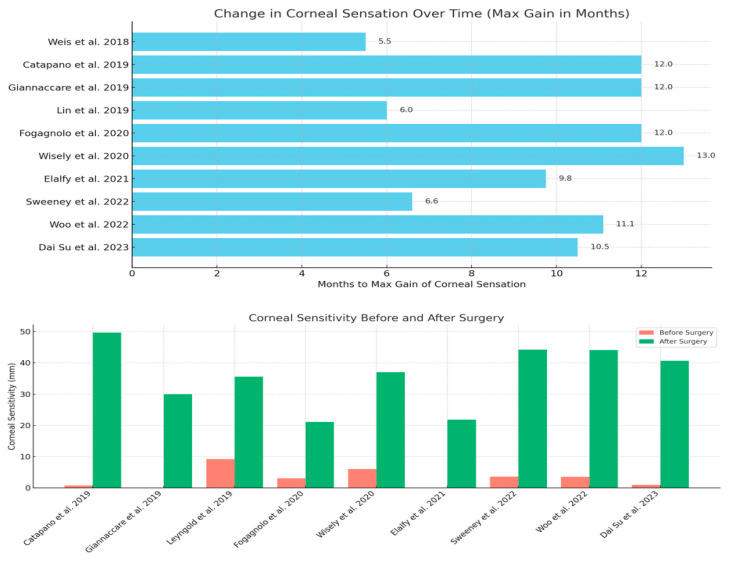
Change in corneal sensation over time. Corneal sensitivity before and after surgery [9,10,13,14,15,16,17,18,19,21,22].

**Figure 4 biomedicines-13-00961-f004:**
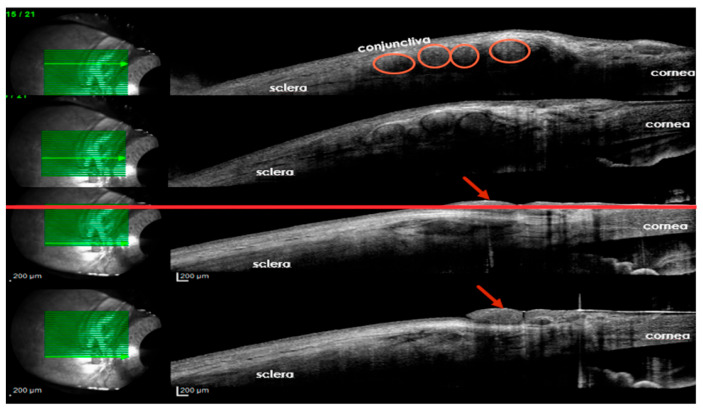
Ocular coherence tomography, 3 years after corneal neurotization, demonstrates the presence of nerve grafts beneath the conjunctiva (circles) at the level of the peripheral cornea (arrows) (personal archive, Samoila Ovidiu).

**Table 1 biomedicines-13-00961-t001:** Clinical studies investigating NT, in chronological order. Study design and outcomes.

Study	No. Eyes/Patients	Age	Sex		Mean Denervation Time (Years)	Type of Surgery (DNT/INT)	Outcomes							Mean Follow-Up (Months)
			♀	♂			Improvement in Corneal Sensation after NT				Visual Acuity	Corneal Epithelial Stability	IVCM	
							Significantly Improved at the End of Follow-Up	Mean Time to First/Max Gain of Corneal Sensation (Months)	Initial CCS	Max CCS				
Weis et al. 2018 [13]	6/6	57	1	5	1.9	INT Sural nerve	Yes 100%	<6 (qualitatively)	NA	NA	83% Improved	All ulcers healed Stable epithelium	NA	12
Catapano et al. 2019 [14]	19/16	12.5	9	7	6.4	INT Sural nerve	Yes 95%	<3/<12	0.8mm	49.7mm (*p* < 0.0001)	Stable	Significantly improved 21% PED after PK	NA	24
Giannaccare et al. 2019 [15]	3/3	60	3	0	3.66	DNT	Yes 100%	<3/12	0 mm	30 mm	Stable	All healed <3 months	SNP+ at 3 months, Max at 12 months	12
Leyngold et al. 2019 [10]	7/7	46	3	4	4.3	INT Acellular nerve allograft	Yes 100%	NA	9.2 mm	35.6 mm	Improved overall	All healed	SNP+ at 4 months in 1 case examined	6
Lin et al. 2019 [16]	13/13	61.8	6	7	15.2	DNT ipsilateral	Yes 100%	NA	NA	NA	Improved overall	77% healed, 1 PED	SNP+ at 6 months	18.5
Fogagnolo et al. 2020 [17]	26/25	45.44	20	5	4.7	DNT-16 INT-10 Sural nerve	Yes 80% DNT/ 83% INT	NA	3.07 mm	21.1 mm	Improved overall 0.29 to 0.46 decimal	All healed, 3.9 months	SNP+ at 3 months, normal at 12 months	18.76
Wisely et al. 2020 [18]	5/4	47.4	2	2	1.48	DNT	Yes 100%	<4.2 (2–9)/ NA	6 mm	37 mm	Improved overall	All healed	SNP+ ant the time of PK (13 months)	15.8
Elalfy et al. 2021 [19]	11/11	43	8	3	NA	INT Sural nerve	Yes 82%	1.5–18/ NA	0 mm	21.8 mm	Improved in 55%,	Significantly improved, 91% had reduction in corneal staining	SNP+ in 5 cases at 3 months, plateau up to 12 months	14.5
Thomson et al. 2021 [20]	11/11	45	8	3	5.75	INT Sural nerve	Yes 73%	NA	0 mm	NA; 60 mm 1 case; 40–50 mm 2 cases; 20–40 mm 2 cases; 10–20 mm 4 cases.	Improved in 89%	Significantly improved, 89% had reduction in corneal staining, no PED	NA	10
Sweeney et al. 2022 [9]	17/17	42.6	9	8	NA	INT Acellular nerve allograft	Yes	3.7/6.6	3.6mm	44.2 mm (*p* < 0.01)	Stable	Significantly improved	NA	17.7
Woo et al. 2022 [21]	28/23	15.6	12	11	6.45	INT Sural nerve	Yes (60.7%)	3/11.1	3.5mm	44.1 mm (*p* < 0.001)	Improved overall	32.1% rate of epithelial break-down	NA	37.8
Dai Su et al. 2023 [22]	18/18	34.83	15	3	3.6d	INT Sural nerve	Yes <100%	6/12-18	1.mm	40.63 mm (*p* < 0.0001)	NA	42.9% healed at 6 months, 100% healed at 12 months	SNP+ at 6 months, max at 12–18 months	24

NA—not available; INT—indirect neurotization, DNT – direct neurotization; NT—neurotization; CCS—central corneal sensibility (on CBA); IVCM—in vivo confocal microscopy; PED—persistent epithelial defect; PK—penetrating keratoplasty; SNP—corneal sub-basal nerve plexus. Statistical significance was noted if present (*p* < 0.05)—MaxCCS versus initial CCS.

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
