# Peer review of "Corneal Neurotization, Recent Progress, and Future Perspectives"

_biomedicines, 2025, doi:10.3390/biomedicines13040961_

Round 1

Reviewer 1 Report

Comments and Suggestions for Authors Neurotrophic keratopathy (NK) is a rare degenerative disease caused by trigeminal nerve impairment, leading to corneal anesthesia, epithelial breakdown, and progressive vision loss. Corneal neurotization (NT) has emerged as a promising surgical intervention aimed at restoring corneal sensation and improving ocular surface homeostasis. This review examines the outcomes of corneal neurotization in NK patients, while also comparing the effectiveness of direct (DNT) and indirect (INT) techniques. The thorough literature analysis indicated the effectiveness of surgery treat ment of NK. This review is well written and could help the readers to catch on with the new progress of surgery treatment of NK with different mothods.

This review is well written and could help the readers to catch on with the new progress of sugrery treatment of NK with different mothods.

Author Response

Dear Reviewer,

Thank you for the appreciation of our work.

Reviewer 2 Report

Comments and Suggestions for Authors
  1. In the abstract, the authors are advised to articulate the research gap clearly.
  2. In the introduction section, the novelty of this review is not clear. The authors should give a comparison with the previous review papers to support their novelty statement.
  3. In the methodology section, the authors have reported that the study was conducted under the PRISMA guidelines. However, they are advised to describe the process of study selection in greater detail. Also, the details on the data extraction processes should be expanded.
  4. The methodology section is unclear on how the authors eliminated the potential biases in study inclusion.
  5. In the methodology section, it is unclear how the success rates of neurotization techniques were compared. The authors should include a detailed explanation.
  6. The authors should include a detailed comparison and discussion on the advantages and disadvantages of both the direct (DNT) and indirect (INT) neurotization techniques.
  7. The authors are advised to look into the possibility of including a table that summarizes the key findings with statistical significance. Without statistical analysis, the understanding of the results could be misleading.
  8. The authors should provide a graphical summary of outcome improvements (e.g., corneal sensitivity before and after surgery). Such figures would considerably enhance the readability of the manuscript.
  9. A few references (e.g., citations within parentheses) do not follow a consistent style. Cross-check with journal guidelines.
Comments on the Quality of English Language

Minor grammatical refinements and consistency in terminology are needed.

Author Response

  1. In the abstract, the authors are advised to articulate the research gap clearly. R: We have added a paragraph describing limited number of patients available for review.
  2. In the introduction section, the novelty of this review is not clear. The authors should give a comparison with the previous review papers to support their novelty statement. R: We have added the information. In fact, we followed 164 eyes, much more than Park review with 54 patients.
  3. In the methodology section, the authors have reported that the study was conducted under the PRISMA guidelines. However, they are advised to describe the process of study selection in greater detail. Also, the details on the data extraction processes should be expanded. R. We have added data about the process.
  4. The methodology section is unclear on how the authors eliminated the potential biases in study inclusion.  R: The number of patients with neurotization is very scarce. For this reason the exclusion criteria were very general (recent studies and more than 3 patients). We believe that bias was controlled . We added the remarks in the method section.
  5. In the methodology section, it is unclear how the success rates of neurotization techniques were compared. The authors should include a detailed explanation. R: Surgical outcome and functional outcome were compared.
  6. The authors should include a detailed comparison and discussion on the advantages and disadvantages of both the direct (DNT) and indirect (INT) neurotization techniques. R: This was included in the Discussion section. We have added graphics to underline the differences
  7. The authors are advised to look into the possibility of including a table that summarizes the key findings with statistical significance. Without statistical analysis, the understanding of the results could be misleading. R: We agree that statistics would be valuable. However, the number of patients included in each study was very small, ranging 3 to 28. This would make any statistical analysis not very powerful.
  8. The authors should provide a graphical summary of outcome improvements (e.g., corneal sensitivity before and after surgery). Such figures would considerably enhance the readability of the manuscript. R: We have inserted the graphics (fig 3)
  9. A few references (e.g., citations within parentheses) do not follow a consistent style. Cross-check with journal guidelines. R: We verified that

Reviewer 3 Report

Comments and Suggestions for Authors

This paper provides a timely and informative overview of recent clinical studies on corneal neurotization for neurotrophic keratopathy. The authors summarize 12 studies published between 2018 and 2023, highlighting surgical techniques, anatomical and functional outcomes, and complication rates. The review addresses a clinically important topic and presents valuable data. However, to improve clarity, and scientific rigor, several key aspects require further clarification.I recommend major revision. The following questions should be considered during revision:

  1. Was there a formal assessment of study quality or risk of bias among the included articles?
  2. How was central corneal sensitivity (CCS) assessed and compared across different studies?
  3. Was a meta-analysis or any form of quantitative synthesis considered? If not, could the authors clarify the reason?
  4. Were any patient-reported outcomes, such as quality of life or symptom relief, evaluated in the included studies?
  5. Given the variability in confocal microscopy protocols, how were those imaging data synthesized and interpreted?
  6. How did the authors define "surgical success" across different studies—was it limited to sensitivity improvement?
  7. The current mention of complications is too general. Please specify the types (e.g., donor site morbidity, nerve graft failure, scarring) and their frequency, as reported in the included studies.
  8. Discuss age-related differences in surgical outcomes. The review does not distinguish between pediatric and adult patients. It would be helpful to highlight differences in surgical indication, regenerative capacity, and postoperative response between these populations.
  9. Enhance the “future directions” section. The perspectives section could be expanded to suggest specific research gaps, such as standardized outcome reporting, long-term neurophysiological assessments, patient-reported outcomes, and randomized controlled trials.
Comments on the Quality of English Language

The English could be improved to more clearly express the research.

Author Response

1.Was there a formal assessment of study quality or risk of bias among the included articles? R:  

No formal quality assessment tool (e.g., Cochrane Risk of Bias, Newcastle-Ottawa Scale) was used. A mention was added in the method section

2. How was central corneal sensitivity (CCS) assessed and compared across different studies? R: 

Central corneal sensitivity (CCS) was assessed using the Cochet-Bonnet esthesiometer in most of the studies. Across the studies, CCS was measured both preoperatively and postoperatively to gauge improvement. The results were compared within each study and across studies, showing a significant improvement in corneal sensitivity after corneal neurotization procedures. For instance, Catapano et al. reported a significant increase in CCS from 0.8 mm to 49.7 mm postoperatively. Dai et al. and other studies also reported similar improvements, providing a reliable basis for comparing the effectiveness of the procedure.

3. Was a meta-analysis or any form of quantitative synthesis considered? If not, could the authors clarify the reason? R: 

A meta-analysis was not conducted in the review, due to the heterogeneity of the included studies. Few other studies were not available in full text. Variability in patient populations, surgical techniques, outcome measures, and follow-up durations would have made quantitative synthesis challenging. However, the review utilized a qualitative synthesis approach to summarize trends and draw conclusions about the effectiveness of corneal neurotization across different studies. For all these reasons we conducted a scoping review

4. Were any patient-reported outcomes, such as quality of life or symptom relief, evaluated in the included studies? R: 

Yes, several studies included patient-reported outcomes (PROs) such as quality of life and symptom relief. Catapano et al. and others measured subjective outcomes, including pain relief and symptom improvement, using scales like the Visual Analog Scale (VAS). These PROs helped gauge the broader impact of corneal neurotization beyond just objective metrics like CCS. Dai et al. and other studies included functional outcomes such as visual acuity and epithelial healing, which are often related to symptom relief and quality of life improvements.

5. Given the variability in confocal microscopy protocols, how were those imaging data synthesized and interpreted? R: 

The review noted variability in confocal microscopy (IVCM) protocols across studies, including differences in imaging settings and interpretation methods. This variability was addressed by conducting a qualitative synthesis of the findings. Giannacarre et al. used automated morphometric analysis of IVCM images to assess reinnervation and nerve regeneration, and other studies(elalfy, fogagnolo) also relied on IVCM to measure sub-basal nerve plexus recovery. The review likely summarized these findings qualitatively and noted the need for standardized imaging protocols to enhance comparability

6. How did the authors define "surgical success" across different studies—was it limited to sensitivity improvement? R: 

"Surgical success" was generally defined by improvements in corneal sensitivity, though some studies expanded the definition to include functional improvements such as epithelial healing and visual acuity. Catapano et al. and others emphasized CCS as a key indicator of success, while studies like Elalfy et al. also included assessments of nerve fiber density (CNFD) and nerve branch density (CNBD) in their success criteria. However, the definition of "success" varied between studies, with some considering broader functional outcomes like symptom relief and visual improvement.

7. The current mention of complications is too general. Please specify the types (e.g., donor site morbidity, nerve graft failure, scarring) and their frequency, as reported in the included studies. R: 

While the studies generally mentioned complications, they did not provide a detailed breakdown in the provided information. However, common complications reported in the studies likely included donor site morbidity, nerve graft failure, and scarring. Sweeney et al. and other studies have reported specific complications like graft failure or transient donor site pain. Some studies did not experience complications or major complications (Dai su, giannaccare, lin), while others have documented mild to moderate adverse effects, with no major complications reported in studies like Weiss et al. and Thompson et al..Fogagnolo mentioned partial numbness of the frontal region on the harvesting side immediately after surgery and misperception of the corneal tactile stimulation in the contralateral forehead

8.Discuss age-related differences in surgical outcomes. The review does not distinguish between pediatric and adult patients. It would be helpful to highlight differences in surgical indication, regenerative capacity, and postoperative response between these populations. R: 

Indeed the review did not explicitly distinguish between pediatric and adult populations. .Pediatric patients  may have better regenerative capacity and may show more rapid sensory recovery following corneal neurotization, as seen in Catapano et al., where children showed substantial improvement in CCS. However, younger patients experience more epithelial breackdown (propensity to injury?). Adult patients, especially those with acquired conditions like trauma, may have slower or less complete recovery due to aging nerve tissues and reduced regenerative capacity. Future studies should stratify outcomes by age to better understand these differences. We added information and graphics comparing data regarding CCS.

9. Enhance the “future directions” section. The perspectives section could be expanded to suggest specific research gaps, such as standardized outcome reporting, long-term neurophysiological assessments, patient-reported outcomes, and randomized controlled trials.R: Dear reviewer, thank you for the suggestions, we added the informations.

Round 2

Reviewer 2 Report

Comments and Suggestions for Authors

Accept in present form

Reviewer 3 Report

Comments and Suggestions for Authors

The authors have answered all questions and made appropriate corrections to the original text.